# Cerebral Malaria Model Applying Human Brain Organoids

**DOI:** 10.3390/cells12070984

**Published:** 2023-03-23

**Authors:** Rita Silva-Pedrosa, Jonas Campos, Aline Marie Fernandes, Miguel Silva, Carla Calçada, Ana Marote, Olga Martinho, Maria Isabel Veiga, Ligia R. Rodrigues, António José Salgado, Pedro Eduardo Ferreira

**Affiliations:** 1Life and Health Sciences Research Institute (ICVS), School of Medicine, University of Minho, Campus Gualtar, 4710-057 Braga, Portugal; 2ICVS/3B’s-PT Government Associate Laboratory, 4710-057 Braga, Portugal; 3CEB—Centre of Biological Engineering, Universidade do Minho, Campus de Gualtar, 4710-057 Braga, Portugal; 4Department of Experimental Biology, Section of Microbiology, Faculty of Science, Masaryk University, Kamenice 753/5, 62500 Brno, Czech Republic; 5LABBELS—Associate Laboratory, 4710-057 Braga, Portugal

**Keywords:** cerebral malaria, brain organoids, transcriptome, human iPSCs, secretome, HBMEC activation

## Abstract

Neural injuries in cerebral malaria patients are a significant cause of morbidity and mortality. Nevertheless, a comprehensive research approach to study this issue is lacking, so herein we propose an in vitro system to study human cerebral malaria using cellular approaches. Our first goal was to establish a cellular system to identify the molecular alterations in human brain vasculature cells that resemble the blood–brain barrier (BBB) in cerebral malaria (CM). Through transcriptomic analysis, we characterized specific gene expression profiles in human brain microvascular endothelial cells (HBMEC) activated by the *Plasmodium falciparum* parasites. We also suggest potential new genes related to parasitic activation. Then, we studied its impact at brain level after *Plasmodium falciparum* endothelial activation to gain a deeper understanding of the physiological mechanisms underlying CM. For that, the impact of HBMEC-*P. falciparum*-activated secretomes was evaluated in human brain organoids. Our results support the reliability of in vitro cellular models developed to mimic CM in several aspects. These systems can be of extreme importance to investigate the factors (parasitological and host) influencing CM, contributing to a molecular understanding of pathogenesis, brain injury, and dysfunction.

## 1. Introduction

Malaria is a parasitic infectious disease that had 247 million reported cases and 619,000 reported deaths worldwide in 2021 [1]. The main *Plasmodium* species that infect humans are *P. falciparum* and *P. vivax.* The most common clinical signs of malaria are anemia and fever, but the most severe form is cerebral malaria (CM), which occurs when the parasite affects the blood–brain barrier (BBB) integrity and triggers an inflammatory response, leading to brain swelling. This can result in neurological sequelae, cognitive deficits, epilepsy, and behavioral difficulties [2].

The BBB is a specialized layer composed of different cell types, including endothelial cells that cover the inner surface of blood vessels in the brain and regulate blood flow, preventing harmful substances from entering. Moreover, endothelial cells also regulate hemostatic function and coagulation through adhesion molecules, such as intercellular adhesion molecule (ICAM), vascular cell adhesion molecule (VCAM), platelet endothelial cell adhesion molecule 1 (PECAM-1), E-selectin, cluster of differentiation 36 (CD36), or endothelial protein C receptor (EPCR) [3,4,5,6].

Endothelial activation is associated with CM and is involved in the pathogenesis and progression of malaria as the endothelium serves as a barrier between the brain and the *P. falciparum*-infected red blood cells (iRBCs) [7]. The adhesion receptors on the cell surface that are involved in sequestration mechanisms of the malaria parasite are associated with microvascular obstruction and clinical complications. In addition, other host receptors, such as EPCR, ICAM-1, and CD36, are involved in the sequestration of iRBCs to endothelial cells [8,9,10,11,12], as well as the *P. falciparum* erythrocyte membrane protein 1 (PfEMP1) [8,13,14,15,16,17] proteins of the parasites in the trophozoite and schizont stages [18].

The precise mechanisms underlying the interaction between human brain endothelial cells and the malarial parasite are not fully understood, but key factors have been identified, including inflammation, vascular leakage, and the uniqueness of *P. falciparum* variants, which are the mechanisms associated with CM [13,19,20]. Upon infection, pro-inflammatory cytokines are released into the blood, activating coagulation and inflammation factors. Disorders in the coagulation system are frequently found in patients with malaria infection because parasitized red blood cells and activated platelets induce amplification of the coagulation cascade. The coagulation system plays an essential role in the sequestration of iRBCs in the cerebral vasculature, which can cause thrombosis of blood microvessels and is also interconnected with the innate immune response as a host defense mechanism. Altered plasma levels of biomarkers, such as thrombomodulin, thrombin, and von Willebrand factor, have been reported in *P. falciparum*-infected patients. These coagulation–inflammation mechanisms contribute to hemostatic dysfunction, leading to coma in *P. falciparum* malaria [21,22,23,24,25].

Infection with the malaria parasite can alter the function of brain endothelial cells and BBB permeability, leading to the accumulation of toxic substances in the brain and interfering with normal brain function. These events can induce edema and interfere with the normal functioning of neurons, leading to seizures, confusion, and other cognitive and behavioral changes [26].

Human brain organoids generated from induced pluripotent stem cells (iPSCs) are an important model for studying neurological disorders [27,28,29,30,31]. A human brain organoid contains different cell types that can self-assemble into a whole brain and mimics the fetal brain with structured brain regions, such as the cerebral cortex, hind/midbrain ¸ and hypothalamus [32,33,34,35,36]. This allows the study of structural phenotypes and understanding the cellular and molecular mechanisms involved in human brain development, disease-induced neuronal disorders, and cognitive impairments [37,38]. Recently, it was shown to be a powerful tool for understanding the neuropathogenesis of infectious parasitic diseases, including cerebral malaria [39,40].

We established a novel approach to understand human CM and its neurological impact in vitro, using malaria parasites, human endothelial cells, and brain experimental models. Our results confirm that *P. falciparum* parasites can modulate human brain microvascular endothelial activation and stimulate inflammatory mechanisms, altering the transcriptome of endothelial cells. Our findings contributed to a better understanding of the brain injury mechanisms associated with CM in a non-invasive manner. These models are essential for investigating brain development, complexity, and organization and for resembling clinical CM in a non-invasive manner.

## 2. Materials and Methods

### 2.1. Malaria Parasite Culture

The *Plasmodium falciparum* strains Dd2 (MR4, MRA-156), HB3 (MR4, MRA-155), and 3D7 (MR4, MRA-102) were kindly provided by the Malaria Research and Reference Reagent Resource Center (MR4) (Manassas, VA, USA). The *P. falciparum* strains were cultured in 75 cm^2^ coated culture flasks with human red blood cells in 15 mL of RPMI-1640, supplemented with 2 mM L-glutamine, 200 μM hypoxanthine, 0.25 μg/mL gentamycin, 25 mM HEPES, 0.2% NaHCO3, and 0.25% Albumax II. Cultures were maintained at 37 °C in a humid atmosphere of 5% O_2_/5% CO_2_/90% N_2_. Parasite growth was monitored through Giemsa-stained blood smears. Parasite stage synchronization and purification were performed using a magnetic cell separation, AutoMACS (Miltenyi Biotec), according to the manufacturer’s protocol using the program PMalaria. This cell enrichment procedure allows obtaining highly viable and pure parasite culture at late stages of trophozoite/schizonts. The trophozoite stage is when PfEMP1 is expressed on the surface of *P. falciparum*-infected red blood cells (iRBCs) and is the stage used for cytoadherence assays to microvascular endothelial cells [11].

### 2.2. Endothelial Cell Line-Human Brain Microvascular Endothelial Cells

The human brain microvascular endothelial cells (HBMEC) immortalized cell line was proved to be the most suitable human cell line for a BBB model in vitro, in a monoculture system, due to the tightness of the barrier [41]. The HBMEC were plated in 75 cm^2^ coated culture flasks, cultured with 15 mL of *Dulbecco’s Modified Eagle Medium* (*DMEM*) (ThermoFisher, Waltham, MA, USA) supplemented with 10% fetal bovine serum (FBS) (ThermoFisher, USA) and 1% PenStrep (100 U/mL). Cultures were incubated at 37 °C, 5% CO_2_, in a humidified atmosphere and split in a ratio of 1:3, once a week, using Accutase (Invitrogen, Thermo Fisher Scientific, USA).

HBMEC stimulation endothelial cells with a confluence of about 80% were washed five times with previously warmed phosphate-buffered saline (PBS) (1X). The conditioned media was created by adding half of the cell culture media (2.5 mL of DMEM supplemented with 1% PenStrep [100 U/mL], without FBS) and half of the parasite culture media (2.5 mL) to the endothelial cell lines. A negative control (NC) media was prepared by mixing HBMEC culture with the endothelial cell media and parasite culture media, without parasites. The *P. falciparum* parasite wild-type strains HB3, 3D7, and Dd2 (±12 million parasitized red blood cells) in trophozoite/schizont forms were added to endothelial cell lines and incubated for 4 h and 24 h. Tumor necrosis factor-alpha (TNF-α) [20 ng/mL] (ThermoFisher, USA) was added to the endothelial cells in 25 cm^2^ coated culture flasks or 6-well cell plates (SPL Life Sciences, Republic of Korea) at the same time points and used as a positive control. The media used to culture endothelial cells were collected (secretome) after 4 h and 24 h of stimulation and centrifuged for 5 min at 1200× *g* rpm. The secretome was collected until a 500 μL volume was left and then stored at −80 °C.

### 2.3. Flow Cytometry of HBMEC Cells

For flow cytometry, the 6-well cell plates (catalogue number 30006, SPL Life Sciences, Pochon, Republic of Korea) containing cells from each condition previously described in the methodology section “HBMEC stimulation” were washed three times with warmed PBS. Accutase^®^ (ThermoFisher, USA) was added to the flasks to cover the cells and kept at room temperature (RT) for 5 to 10 min. The cells were detached, washed with fluorescence-activated cell sorting (FACS buffer), and then centrifuged at 1200× *g* rpm at 4 °C for 2 min. The cells were then stained with cocktail antibodies diluted in FACS buffer for 20 min in the dark at RT. After that, the cells were washed twice with FACS buffer and centrifuged (2 min, 1200× *g* rpm, 4 °C), and the supernatant was discarded. The cells were then fixed in PBS containing 4% formaldehyde. Flow cytometry analysis was carried out using a FACS Calibur with the CellQuest Pro software (BD Biosciences) to screen the endothelial cell surface receptors expression involved in *P. falciparum* adhesion. The following antibodies were used: anti-ICAM-1, anti-VCAM-1, antiCD31/PECAM-1 (Antibodies-Online, Aachen, Germany), anti-CD36 (Abbexa, Cambridge, UK), anti-CD201/PROCR/EPCR (Asssaypro, St Charles, MO, USA), anti-ICAM-2 (Thermofisher Scientific, USA), anti-CD 325/N-cadherin (Thermofisher Scientific, USA), CD321/JAM-A (Thermofisher Scientific, USA), CD51/61/Integrin-αVβ3 (Thermofisher Scientific, USA), goat anti-rabbit IgG (Thermofisher Scientific, USA), and goat anti-mouse (Thermofisher Scientific, USA).

The data were obtained from five independent experiments, each with three replicates. The data analysis was performed using Graphpad Prism version 7.0, and one-way and two-way ANOVA tests were applied, with Tukey’s posttest and Bonferroni post hoc test, respectively.

### 2.4. RNA Extraction of HBMEC Cells

HBMEC cells were obtained from 25 cm^2^ coated culture flasks, and three independent experiments were conducted. The results from these experiments were combined for each HBMEC condition. RNA extraction was carried out using the RNeasy^®^ Mini kit (QIAGEN, Hilden, Germany). To prevent genomic DNA contamination, RNase-free DNase I (QIAGEN, Germany) was used as per the manufacturer’s instructions. The RNA was dissolved in sterile water and assessed spectrophotometrically using Nanodrop for quantification.

### 2.5. RNA-Seq Analysis of P. falciparum Activated HBMEC Cells

Total RNA was sequenced and analyzed for mRNA expression profiles by Macrogen, South Korea. The RNA quality control (QC) was performed, and only qualified samples were used for library construction. The library was constructed by randomly fragmenting the DNA/cDNA sample, followed by 5′ and 3′ adapter ligation.

Trimmed reads were mapped to the reference genome using HISAT2, which is a splice-aware aligner; then a transcript was assembled with the aligned reads. The expression profiles were represented as reading count and normalization value based on transcript length and depth of coverage. The normalization value used was either FPKM (Fragments Per Kilobase of transcript per Million Mapped reads) value or RPKM (Reads Per Kilobase of transcript per Million mapped reads). Statistical hypothesis testing was used to filter out groups with different conditions, genes, or transcripts that express differentially, according to the criteria of fold change >1.5.

### 2.6. HBMEC-P. falciparum Activated Secretome Analysis

The proteins secreted by cells, known as the secretome, play a vital role in biological and physiological processes. To study the impact of erythrocytes infected with different strains of *P. falciparum* on the inflammatory response of HBMEC, we used a human inflammation RayBio^®^ C-Series membrane-based antibody arrays membranes (Raybiotech, Norcross, GA, USA) to analyze secretome composition. HBMEC cells were stimulated with the *P. falciparum* strains 3D7, HB3, and Dd2 for 4 h and 24 h. The resulting secretome was then added to the human inflammation membranes. The membrane-based antibody arrays is based on a biotinylated antibody cocktail and HRP-Streptavidin, allowing for the semi-quantitative detection of 40 human proteins including cytokines and chemokines, namely: IL-1*β/IL-1 F2*, IL-1 *α*/IL-1 F1, IL-2, IL-3, IL-6, IL-6R, IL-8/CXCL8, IL-12p40, IL-12p70, IL-15, IL-16, IL-17, IL-17A, GM-CSF, M-CSF, IFN-*γ*, MCP-1/CCL2, MIP-1*α*/CCL3, MIP-1*β*/CCL4, RANTES/CCL5, TGF β1,TNF-α, TNF RI/TNFRSF 1A, TNF β1/TNFSF1B, TNF RII/TNFRSF 1B, MIP-1 DELTA/CCL15, MIG/CXCL9, IP-10/CXCL10, IL-4, IL-10, IL-11, IL-13, GCSF, Eotaxin-1/CCL11, Eotaxin-2/CCL24/MPIF-2, TIMP-2, IL-7, IL-8/CXCL8, PDGF-BB, MCP2/CCL8, M-CSF, ICAM-1/CD54, and I309/TCA/CCL1. Chemiluminescence detection was used as an imaging system, in the Sapphire Molecular Imager. Data were obtained from two replicates in each condition and normalized to the internal control signal of each protein array. Data analysis was performed with Graphpad Prism version 7.0 and applied Two-way ANOVA using Dunnett’s multiple comparisons test to compare the expression level of proteins in the secretome between the conditions with stimulation (3D7, HB3, Dd2, and TNF-α) and NC.

### 2.7. iPSCs Culture and Human Brain Organoid Differentiation Media

Human iPSCs, by Marote et al. [42], were used to generate the brain organoids. iPSCs were cultured on Vitronectin XF™ (Stem Cell Technology, France) coated plates in mTeSR™1 medium (Stem Cell Technology, France), kept at 37 °C (5% CO_2_) for 6–7 days with daily medium changes, and split using manual passage. The generation of human brain organoids was based on protocol described in Lancaster et al., 2014 [30]. To initiate the differentiation, embryoid bodies (EBs) were formed by detaching the iPSCs with TrypLE™ Express Enzyme (ThermoFisher). Then, cells were plated into a low attachment 96-well plate (Thermo Fischer) in DMEM/12 Advanced supplemented with 15% Knockout serum replacement (KSR) (Thermo Fischer), 1% non-essential amino acids, 2% glutamax (Thermo Fischer), 2-mercaptoethanol (55 mM, Thermo Fischer), and Rho-associated protein kinases inhibitor (Y-27632, 5 mM—StemCell Technology). After 6 days, we initiated neural differentiation by culturing the spheres in DMEM/12 Advanced medium supplemented with 1% non-essential amino acids, 1% glutamax, and 1% of N_2_ (1x) supplement (Thermo Fischer) and heparin (1 mg/mL, Sigma-Aldrich, St. Louis, MO, USA). After 5 days, the neurospheres were embedded in Geltrex™ LDEV-Free hESC-qualified Reduced Growth Factor Basement Membrane Matrix (Thermo Fischer) for at least 40 min. Then the culture was switched to differentiation media [DMEM/F12: Neurobasal (1:1, both from ThermoFisher)], supplemented with 0.5% of N_2_ (1x) supplement, 1% non-essential amino acids, 1% glutamax, 2-mercaptoethanol, B27 1x supplement (Thermo Fischer), and insulin (2.5 mg/mL, Sigma). During 40 days of differentiation, the brain organoids grew with regular brain organoid media.

### 2.8. Human Brain Organoids Stimulation with HBMEC Secretome

After 40 days of differentiation, brain organoids were treated for 5 days with secretome, previously described, and HBMEC stimulation was used to stimulate the brain organoids. Half of the media of brain organoids and half of the HBMEC secretome were used for stimulation and changed every 2 days. The media was collected and stored at −80 °C. Nine secretome conditions were established: positive control (BO_TNF), with TFN-α stimulation at 4 h; three different wild-type strains (3D7, Dd2, HB3) of P. falciparum parasites induction for 4 h (BO_3D7_4H, BO_Dd2_4H, BO_HB3_4H) and 24 h (BO_3D7_24H, BO_Dd2_24H, BO_HB3_24H), respectively; a condition of using half brain organoids media and half secretome in the previous HBMEC stimulation (BO_MM); and a negative control (BO_control) consisting of brain organoids that grew in the organoid media.

### 2.9. RNA Extraction of Human Brain Organoids

RNA extraction was performed with RNeasy^®^ Mini kit (QIAGEN, Germany), and to avoid genomic DNA contamination, the remaining RNA was treated with Rnase-free Dnase I (QIAGEN, Germany) according to the manufacturer’s instructions. For each experiment, five organoids were pooled, resulting in a total of 15 organoids per condition across three independent experiments. The collected RNA was resuspended in sterile water and submitted for RNA sequencing analysis. The analysis utilized 15 organoids per condition, which were pooled from three independent experiments.

### 2.10. RNA-Seq Analysis of Human Brain Organoids

The RNA-seq library was constructed using the TruSeq Stranded mRNA LT Sample Prep Kit (Illumina, San Diego, CA, USA). Whole transcriptome sequencing of Homo sapiens was preformed to examine different gene expression profiles and perform gene annotation on a set of useful genes based on gene ontology pathway information. To map cDNA fragments obtained from RNA sequencing, we used GRCh37 as a reference genome [43,44,45]. Trimmed reads were mapped to the reference genome using HISAT2, a splice-aware aligner, and Stringtie was used for transcript assembly. The expression profile was calculated for each sample and transcript/gene as reading count, FPKM (Fragment per Kilobase of transcript per Million mapped reads)/RPKM (Reads Per Kilobase of transcript per Million mapped reads), and TPM (Transcripts Per Kilobase Million). The DEG samples were sorted by reading the count value of known genes. For nine samples, if more than one read count value was 0, we did not include it in the analysis. Therefore, out of a total of 35,993 genes, 16,953 were excluded and only 19,040 genes were used for statistical analysis. To reduce systematic bias, we estimated size factors from the read count data using the read count data (calcNormFactors method). The read count data were normalized with the Trimmed mean of the M-values (TMM) method for an exact test in edgeR. Then, a statistical test was performed with the normalized data. The DEG analysis was performed on eight comparison pairs using edgeR and the results showed that 2045 genes satisfied |fc| ≥ 2 and exactTest raw *p*-value.

### 2.11. Human Brain Organoids Cryosection

The brain organoids from each condition were transferred to a tube containing a minimal amount of media. Then, 1 mL of 4% PFA was added and left at 4 °C for 30 min. After aspirating the PFA, a gradient of sucrose was applied. The cryosectioning of the organoids was performed using a sucrose gradient of 10%–20%–30%, each for 30 min, as previously described by Lancaster et al. [30]. The organoids were then kept in 30% sucrose solution overnight at 4 °C. Subsequently, the organoids were embedded in OCT blocks and stored at −80 °C. After freezing, the blocks were placed in a cryostat, and cryosections of 10–15 μm were obtained (Leica, Wetzlar, Germany).

### 2.12. Immunofluorescence of Human Brain Organoids

The slides containing brain organoid cryosections were washed twice with PBS (1x) and blocked with 0,1% TBST with 0.1% BSA diluted in PBS (1x). The following primary antibodies, diluted in 1% BSA, were incubated for 4 h at RT: anti-NESTIN (1:200, mouse—Millipore, Burlington, MA, USA), anti-SOX-2 (1:500, mouse—Biolegend, San Diego, CA, USA), anti-TBR1 (1:200, rabbit—Sigma Prestige, St. Louis, MO, USA), anti-β Tubulin III (1:1000, mouse—Promega, Madison, WI, USA), anti-DCX (1:1000, rabbit—Abcam, Cambridge, UK), anti-MAP-2 (1:500, mouse—Abcam), anti-GFAP (1:200, mouse—DAKO, Glostrup, Denmark), and anti-cleaved caspase-3 (1:100, rabbit—Millipore). The cryosections slides were washed twice with PBS (1X). The following secondary antibodies (1:1000, all from ThermoFisher) diluted in BSA (1%) were incubated for 2 h at RT: AlexaFluor 488 goat anti-mouse and AlexaFluor 594 goat anti-rabbit. The cryosections slides were washed twice with PBS (1x), followed by immunostaining with DAPI (1:1000) diluted in PBS (1x) for 30 min at RT. The cryosections slides were washed twice with PBS (1x), mounted with a mounting media, and stored at 4 °C.

## 3. Results

The transcriptome of HBMEC was analyzed after stimulation with three different *P. falciparum* cell lines (3D7, Dd2, HB3) for 4 h and 24 h, along with NC and positive control.

The total human transcriptome was analyzed after quality control (Appendix A). Multidimensional analysis (Figure 1A) and an analysis of differentially expressed genes (DEG) (Figure 1B) showed that the 4 h activation was highly similar among analyzed samples, as were the transcriptomes obtained for 24 h activation. All experimental transcriptomes clustered as expected. According to Pearson’s coefficient, the NC and TNF-α 24 h were the most similar conditions (*p* = 0.99), while 3D7 and TNF-α were the most dissimilar (*p* = 0.93) (Appendix A). Furthermore, hierarchical clustering analysis (Appendix A) identified 8005 genes that satisfied the |fc| ≥ 2 condition and all conditions at 4 h activation share a similar expression profile (Appendix A). The effect of TNF-α on HBMEC cells was diminished at 24 h compared with 4 h, indicating that TNF-α stimulation decays over time [46,47,48,49].

The *P. falciparum* 3D7 strain had the highest impact on HBMEC activation both for up- and down-regulation of transcriptomic gene expression, as compared with other strains (Figure 1C). Additionally, volume plot analysis identified the top five genes (red spots) with altered transcriptomic variation, with at least two-fold differences, as RNR2 and COX1 (up-regulated), and TXNRD1, PLEC, and FLNB (down-regulated) (Figure 1D and Appendix A). At 4 h of *P. falciparum* activation, the PABPC1 gene was up-regulated, and at 24 h, the HSPA8 gene was down-regulated in all conditions (Appendix A). Moreover, it is possible to verify the profiles of prostaglandins, RNR2, COX-2, COX-3, and the reactive oxygen species (ROS) modulator 1 (ROMO1) gene (Appendix A). Our transcriptomic results showed an up-regulation of COX-1 (fc ≥ 2), COX-2 (fc ≥ 4), and COX-3 (fc ≥ 4.5) gene expression for all conditions, except for TNF-α at 24 h activation (Appendix A). It is possible to verify in our transcriptomic results that the prostaglandin E receptor 1 (PTGER1) gene is up-regulated (|fc| ≥ 3) by *P. falciparum* 3D7 strain at 4 h and 24 h activations (Appendix A). Our transcriptomic results also showed an increased expression of RNR2 and RNR1 (Appendix A).

### 3.1. Transcriptomic Evaluation of Novel Receptor Genes Activated by P. falciparum

We have identified the activation of a novel endothelial adhesion receptor gene, *ICAM-2*, with up-regulation (fc ≥ 3) across all conditions tested (Appendix A). Specifically, the 3D7 strain showed up-regulation (fc ≥ 2) of the *F11R*/*JAM-1* gene, which encodes the JAM-A protein, at 24 h activation. The *JAM-2* gene, which encodes the JAM-B protein, also showed up-regulation (fc ≥ 2) at 24 h by the *P. falciparum* HB3 strain (Appendix A). Furthermore, the 3D7 strain at 4 h activation showed specific up-regulation of the *ICAM-3* and *ICAM-4* genes, both with fc ≥ 2. TNF-α (4 h) led to a down-regulation (fc ≥ 2), as did also the Dd2 strain (fc ≥ 2) and HB3 at 24 h (fc ≥ 5), of *ICAM-4* (Appendix A). However, the expression of the *ICAM-1* gene did not show relevant fold-change alteration (Appendix A). At both 4 h and 24 h of activation, our transcriptomes showed an up-regulation of *PECAM-1* (fc ≥ 2) for all *P. falciparum* strains, independent of activation time, except for Dd2 at 4 h activation (Appendix A). The *Integrin αV* gene showed a similar profile as *PECAM-1* with an up-regulation for all conditions, except for TNF at 24 h and HB3 at 24 h of activation. Moreover, the HBMEC transcriptomes showed an up-regulation (fc ≥ 2) of *N-cadherin* for 3D7 at 4 h and 24 h and HB3 at 4 h (Appendix A).

### 3.2. Receptors Expression of HBMEC by P. falciparum Strains

We assessed the activation of classic receptors linked to *P. falciparum* adhesion to HBMEC. The presence of the pro-inflammatory cytokine TNF-α resulted in a significant up-regulation of endothelial markers PECAM-1 (1.189; ±0.063; *p* ≤ 0.05) and ICAM-1 (1.349; ±0.141; *p* ≤ 0.05) after 24 h of activation and PECAM-1 (1.104; ±0.034; *p* ≤ 0.05) for the shorter activation time (4 h) (Figure 2), in line with previous reports [50]. However, no significant differences were observed in the levels of other surface proteins.

The VCAM-1 (Figure 2B) and ICAM-1(Figure 2C) receptors did not show statistically significant changes in expression under *P. falciparum* conditions.

The results showed an increase in the CD36 receptor over the activation time, with an upregulation at 4 h only in the presence of the 3D7 strain (1302; ±0.098; *p* ≤ 0.05). However, at 24, the expression levels of this surface receptor significantly increased in the presence of 3D7 (1.795; ±0.192; *p* ≤ 0.0001), Dd2 (1.513; ±0.081; *p* ≤ 0.01), and HB3 (1.445; ±0.088; *p* ≤ 0.01) (Figure 2E).

The EPCR receptor showed a significant increase in the presence of the 3D7 parasites (1.461; ±0.142; *p* < 0.01) at 24 h (Figure 2F).

The endothelial N-cadherin receptor exhibited a down-regulation in the presence of *P. falciparum* Dd2 (0.933; ±0.026; *p* ≤ 0.05) at 4 h (Figure 3). In the presence of 3D7 parasites at 24 h, JAM-A (1.153; ±0.051; *p* ≤ 0.05) and Integrin αVβ3 (1.373; ±0.125; *p* ≤ 0.05) receptors showed an up-regulation profile. However, at 4 h of activation, all parasite strains exhibited a down-regulation with a similar trend for IαVβ3 expression in 3D7 (0.889; ±0.032; *p* ≤ 0.05), Dd2 (0.843; ±0.032; *p* ≤ 0.001), and HB3 (0.898; ±0.0371; *p* ≤ 0.05). On the other hand, an up-regulation was detected for the JAM-A receptor at 24 h in the presence of the *P. falciparum* Dd2 strain (1.228; ±0.061; *p* ≤ 0.001) and HB3 *P. falciparum* strain (1.153; ±0.052; *p* ≤ 0.0001), revealing significant statistical differences. The results are tabulated in Appendix A.

Endothelial receptors, such as PECAM (*p* ≤ 0.01), CD36 (*p* ≤ 0.01), and Integrin αVβ3 (*p* ≤ 0.0001), were over-expressed by *P. falciparum* 3D7 at 24 h of activation (Figure 2J–M). Moreover, a significant increase in JAM-A (*p* ≤ 0.0001) was observed after exposure to the *P. falciparum* HB3 and Dd2 strain (*p* ≤ 0.05) at 24 h. Overall, the results suggest that 24 h activation had a greater impact on HBMEC receptor expression than 4 h (Figure 2J–M).

*P. falciparum* activation of inflammatory, coagulation cascade, TIMPs, and MMP genes from HBMEC. Our transcriptomic results revealed an increase in the expression of the *cxcl11* gene (fc ≥ 3) for all conditions. However, at 24 h of activation, a higher increase in expression was observed with *P. falciparum* 3D7 (fc ≥ 43), Dd2 (fc ≥ 21), and HB3 strains (fc ≥ 6.5) (Appendix A). The *RANTES* gene was up-regulated (fc ≥ 3), particularly at 24 h of activation (fc ≥ 7) by *P. falciparum* strains (Appendix A). At 4 h activation, the *P. falciparum* 3D7 (fc ≥ 34), Dd2 (fc ≥ 5), and HB3 (fc ≥ 29) strains, and at 24 h activation, the 3D7 (fc ≥ 11), Dd2 (fc ≥ 9) and HB3 (fc ≥ 10) strains showed an increased *cxcl4* gene expression profile. However, the most significant evidence of an increase occurred mainly at 4 h of activation. The *cxcr4* gene was up-regulated (fc ≥ 5) for *P. falciparum* conditions.

The *eotaxin-2* gene was up-regulated (fc ≥ 2) in all conditions (Appendix A), except for TNF-α at 24 h. This result agrees with the previous transcriptomics findings, where TNF-α at 24 h did not have a high impact on activating HBMEC. Additionally, both 3D7 and HB3 strains had a more substantial activation impact (fc ≥ 6) on *eotaxin-2* gene expression at 4 h compared to 24 h (Appendix A). There was no substantial change in the *MCP-1/ccl2* expression gene across conditions, except for an increased expression at 24 h activation for *P. falciparum* 3D7 (fc ≥ 2) and Dd2 (fc ≥ 2) strains (Appendix A). These results demonstrate that gene expression depends on the activation time. The *ccl28* gene was up-regulated for 3D7 at 4 h (fc ≥ 4) and 24 h (fc ≥ 3.5), HB3 at 4 h (fc ≥ 2) and 24 h (fc ≥ 3), and Dd2 at 24 h (fc ≥ 2). However, no significant results were obtained for the TNF-α condition, indicating that up-regulation of *ccl28* is specific to *Plasmodium* parasites (Appendix A). The *Cxcl5* gene was up-regulated for 4 h activation (fc ≥ 2) (Appendix A).

Our analysis of the HBMEC transcriptome revealed that the *ANGPTL4* gene was up-regulated when stimulated with *P. falciparum* 3D7 (fc ≥ 30) and HB3 (fc ≥ 20) strains at 4 h. Notably, after 24 h, *ANGPTL4* expression remained elevated (fc ≥ 10) regardless of the strain used. TNF-α expression levels did not have an impact on *ANGPTL4* transcript levels, suggesting that the up-regulation of *ANGPTL4* is a specific response to *P. falciparum* (Appendix A). Coagulation factor III was up-regulated by both *P. falciparum* strains 3D7 and HB3 at 4 h (fc ≥ 10), while the *endothelin-1* gene was specially up-regulated (fc ≥ 10) by *P. falciparum* 3D7 at 4 h (Appendix A). Additionally, the *TSP-1* gene (Thrombospondin-1 protein) was down-regulated (fc ≤ −2) for all conditions at 24 h (Appendix A). The *vWF* and *tropomodulin-2* genes were down-regulated by *P. falciparum* in all conditions (fc ≤ −2), except for TNF-α at 24 h (Appendix A).

*MMP9* is a protease protein-coding gene involved in neuroinflammation and was increased (fc ≥ 3) in response to all *P. falciparum* strains. However, the 3D7 strain at 24 h showed an even higher expression of *MMP9* (fc ≥ 6) and an increased expression of *TIMP-1* (fc ≥ 2) (Appendix A). *TIMP-2* transcriptomic analysis did not yield significant results for *P. falciparum* strain stimulation (Appendix A). The *MMP10* gene was up-regulated (fc ≥ 7) in response to the *P. falciparum* 3D7 strain at 4 h (Appendix A). Conversely, *MMP1* and *MMP3* were down-regulated (fc ≥ −23; fc ≥ −28), respectively, by the 3D7 strain at 24 h (Appendix A). Appendix A summarizes these results.

### 3.3. Impact of P. falciparum Activation in HBMEC Secretome

The pathogenesis of *P. falciparum* CM is influenced by an imbalance of pro- and anti-inflammatory immune responses mediated by cytokines and chemokines, which can be amplified by parasite sequestration [51].

The results of the study showed a down-regulation of RANTES at 4 h for all *P. falciparum* strains, including 3D7 (0.131; ±0.036; *p* ≤ 0.0001), Dd2 (0.168; ±0.070; *p* ≤ 0.0001), and HB3 (0.863; ±0.368; *p* ≤ 0.0001). At 24 h, 3D7 (0.473; ±0.126; *p* ≤ 0.0446) and HB3 (0.725; ±0.270; *p* ≤ 0.0105) strains exhibited under-expression of RANTES (Table 1).

IL-6 was significantly over-expressed in the presence of all three strains, 3D7 (2.068; ±0.344; *p* ≤ 0.0002), Dd2 (2.030; ±0.440; *p* ≤ 0.0001), and HB3 (2.404; ±0.655; *p* ≤ 0.0001), at 4 h. The same profile of secretome activation was observed at 24 h for 3D7 (2.650; ±0698; *p* ≤ 0.0001), Dd2 (2.677; ±0.713; *p* ≤ 0.0001), and HB3 strains (2.025; ±0.429; *p ≤* 0.0001) (Table 1).

The secretome derived from the 3D7 strain showed an increase in MCP-1 (3.123; ±1.072, *p* ≤ 0.0001) and IL-8 (1.992; ±0.531; *p* ≤ 0.0006) expressions at 4 h, and an over-expression of IL-8 (1.938; ±0.519; *p* ≤ 0.0101) at 24 h (Table 1).

PDGG-BB protein levels were significantly higher at 24 h in 3D7 (2.678; ±0.770; *p* ≤ 0.0022) and HB3 (2.174; ±0.698; *p* ≤ 0.0209) strains, and at 4 h in Dd2 (1.297; ±0.286; *p* ≤ 0.0044) strain.

GM-CSF protein levels increased at 24 h with 3D7 (6.049; 1.799; *p* ≤ 0.0001), Dd2 (4.456; ±1.237; *p* ≤ 0.0042), and HB3 (3.532; ±0.688; *p* ≤ 0.0204) strains (Figure 3A). All parasite strains showed an up-regulation of GM-CSF (fc ≥ 2) in the secretome when endothelial cells were stimulated for 24 h (Table 1).

The TIMP-2 protein also showed over-expression in all three strains, 3D7 (2.063; ±0.438; *p* ≤ 0.0005), HB3 (2.338; ±0.505; *p* ≤ 0.0001), and Dd2 (2.197; ±0.633; *p* ≤ 0.0016), at 24 h (Table 1). MCP-1 protein was over-expressed in HB3 (2.459; ±0.738; *p* ≤ 0.0006), 3D7 (2.682; ±0.871; *p* ≤ 0.0006) and Dd2 (2.964; ±0.976; *p* ≤ 0.0001) strains at 24h,.The MCP-1 PDGG-BB, GM-CSF, and TIMP-2 proteins were over-expressed at 24 h, particularly in HB3, 3D7, and Dd2 strains (Table 1).

Human brain organoids development: The human cerebral organoids were generated from iPSCs of healthy adult individuals. After culturing iPSCs colonies (see Appendix A), the differentiation protocol was used to turn the iPSCs into cells from the three germ layers, endoderm, ectoderm, and mesoderm, resulting in the embryoid bodies (EBs) (see Appendix A). Following the differentiation protocol, neural progenitor cells (NPCs) were obtained after 5 days (see Appendix A). These NPCs were then included in Matrigel droplets to create a scaffold that makes it easier to create structures with more complexity. After a few days, neuroepithelial buds developed, and the brain organoids gradually grew more prominent (see Appendix A). To understand the influence of HMBEC-*P. falciparum*-activated secretome in the brain, human brain organoids were challenged with the previously collected secretome. Several conditions were used, including TFN-α stimulation, which was used as a positive control (BO_TNF), and three different wild-type strains of *P. falciparum* parasites (3D7, Dd2, HB3) induced at time points of 4 h (BO_3D7_4H, BO_Dd2_4H, BO_HB3_4H) and 24 h (BO_3D7_24H, BO_Dd2_24H, BO_Hb3_24H), respectively. Brain organoids stimulated with secretome from HBMEC (a mixture of half HBMEC culture media and half *P. falciparum* parasites culture media) called modified media (BO_MM) were also analyzed, as were brain organoids that only grew with brain organoid media (BO_control), which served as a negative control.

The organoids were found to have a large size, with a diameter range of up to 250 µm (see Figure 3A), and exhibited cavities representative of ventricular zones (see Figure 3D). In addition, heterogeneous regions (red square) that resembled choroid plexus architecture and neural tubes (white square) were also found (see Appendix A). The neural microtubule-associated protein 2 (MAP2) marker was expressed throughout the organoids (see Figure 3A), with a higher fluorescence signal forming a neural layer resembling a superficial preplate [32]. MAP-2 is a biomarker of synaptic plasticity and is consistently used to label neurons that have reached a significant level of maturity [52]. Additionally, the newborn neural marker doublecortin (DCX) was also expressed throughout the organoids with a high degree of co-labeling with MAP-2, indicating the presence of active neuronal differentiation within our system (Figure 3A).

The early-born neuron marker transcription factor T-Box Brain 1 (TBR1) was not expressed in the organoid, demonstrating the maturity of the brain organoid (Figure 3B). Positive cells (+) for the neural marker tubulin beta-3 (TUBB3/TUJ1) were found throughout the organoid, denoting the successful differentiation of neuroectodermal tissue. (Figure 3B). The neural stem cell marker NESTIN was expressed in the ventricular zone by NESTIN+ cells (VZ) (see Figure 3C). The glial fibrillary acidic protein (GFAP) astrocytic marker showed a significant population of GFAP+ cells, indicating differentiation along the astroglial lineage (see Figure 3C). In several regions, a substantial degree of GFAP and NESTIN co-labeling was found, indicating the presence of radial-glia/neural-progenitor cells. The marker SRY-Box Transcription Factor 2 (SOX2) of NPC was present, although with a lower staining intensity (Figure 3D). The assessment of caspase 3 (CASP3), which is the molecular executioner of the apoptotic cascade, is crucial for ascertaining the level of cell death within our system. Therefore, a small number of CASP3+ cells could be observed (see Figure 3D). Our immunostaining results demonstrate a reproducible development and maturation of human brain organoids.

BO_Dd2_4H showed an increased presence of the MAP2 marker in the outer zone of brain organoids (Appendix A) compared with the other conditions. The TUBB3 was present in all conditions (see Appendix A). The Nestin marker was present in all conditions; however, at 24 h, there was a reduced presence of the marker in the outer zone of brain organoids, with the BO_TNF condition showing a reduced expression of Nestin (see Appendix A). The SOX2 marker showed a meager presence in all brain organoid conditions studied, and the CASP3 marker was sparsely present in all conditions except for BO_MM (see Appendix A).

GFAP was highly expressed in the conditions of BO_MM, BO_TNF, BO_3D7_4H, and BO_DD2_4H (see Appendix A).

Gene-Enrichment and Functional Analysis Description of Brain Organoids Transcriptome: After normalizing the transcriptomic profiles (Appendix A), it was possible to observe a high similarity between the conditions BO_3D7_24H, BO_Dd2_24H, and BO_HB3_24H (*p* = 0.99) (Figure 4A). The least similar transcriptomes were BO_3D7_4H and BO_TNF (*p* = 0.96), indicating altered effects on brain organoids when challenged with different HBMEC-activated secretomes (Figure 4A). The differential gene expression (DEG) analysis showed 2045 genes’ fold-change (*|fc|* ≥ 2; *p* < 0.05) (see Figure 4B). Down-regulation indicated a normal development slowdown of brain organoids induced by *Plasmodium falciparum* HBMEC-activated secretomes. A multidimensional scaling analysis (MDS) corroborated the previous findings (Figure 4C). The conditions BO_TNF (624 genes), BO_Hb3_4H (589 genes), and BO_Dd2_4H (565 genes) (see Figure 4D) were observed to have a greater influence on the down-regulation of brain organoids genes’ expression Additionally, it was observed that in the BO_MM condition, there was an increased up-regulation of genes compared to other conditions. These findings suggest that the secretome in contact with *P. falciparum* possesses some components that induce the down-regulation of the brain organoids transcriptome.

The results of the biological process enrichment (see Figure 5A) confirmed that the transcriptional signature differed among different conditions, especially for genes related to microtubule bundle, cilium organization, and extracellular and external organization, particularly in the BO_Dd2_4H and BO_HB3_4H conditions compared to the others. The cellular component (Figure 5B) showed a similar profile in the gene signature of microtubule bundle formation and cilium organization for the BO_Dd2_4H and BO_HB_4H conditions. In the cellular component, the extracellular matrix, external encapsulating structure, and collagen-containing extracellular matrix genes had similar gene signatures for most conditions, except for BO_Dd2_4H, BO_HB3_4H, and BO_TNF. The molecular function revealed that the gene signature related to transporter activity was only affected by the BO_TNF condition. The conditions BO_Dd2 4H, BO_HB3 4H, d BO_ TNF, and BO_3D7 24H showed higher down-regulation (Appendix A).

## 4. Discussion

To fulfill the need for a human in vitro system to study CM, we have assembled a collection of cellular technologies to recreate this pathology in the laboratory. In general, the activation of HBMEC by the parasite *P. falciparum* is modulated by the parasite’s genetic background and time-to-activation of HBMEC secretomes. Furthermore, the *P. falciparum* 3D7 4 h condition induces the most significant transcriptomic changes, leading to a higher up- and down-regulation in HBMEC compared to a longer activation period (24 h).

Additionally, the genes *RNR2* and *COX1* showed an up-regulation, while the genes *TXNRD1*, *PLEC*, and *FLNB* were down-regulated (Figure 1D). The production of prostaglandins (PG) from arachidonic acid is catalyzed by the cyclooxygenase (COX) enzymes. Increased expression of COX-1 and COX-2 has been detected in brains of individuals with CM [53,54], and COX inhibitors have shown a protective effect in CM models [54]. The increased presence of COX in CM brains is also associated with higher levels of prostaglandins such as PGE2, which have been linked to fever and inflammation in human CM [53,54]. This research demonstrated that COX1 is one of the genes most expressed in human brain microvascular endothelial cells when they are activated by *P. falciparum*. This establishes a potential link between malaria’s effect on COX-1 gene expression and COX-2 and COX-3 genes.

The *RNR2* gene involves DNA synthesis and repair [55], and cyclic vomiting syndrome disease [56]. The mitochondrial *(MT)-RNR2* upregulation is associated with psychopathy [57]. Although until now no relation has been established between malaria and the *RNR2* gene, our data point to an essential impact of *P. falciparum* on the expression of *RNR2*.

The absence of plectin in endothelial cells has been linked to changes in the cytoskeleton, disruptions in adherent junction, modifications in tight junction modifications, and increased contractility [58,59]. Plectin is also known to play a role in signaling processes in the CNS, and has been associated with various functions in the brain and spinal cord [60] However, there is currently no established connection between the plectin gene or protein and CM. CM has been shown to cause alterations in behavior, neurological function, and cognitive abilities, affecting regions of the CNS and leading to seizures, epilepsy, and other neural disorders [61]. Our results showed a down-regulation in the expression of the plectin gene. Given the role of plectin in CNS signaling and the association between CM and neural disorders, our findings suggest that plectin may be a promising target for further research to better understand its role in CM. The impact of signaling processes in the CNS on neurons and brain and spinal cord functions is known [59]. Actin-binding protein filamin B (FLNB) is predominantly expressed in endothelial cells and plays a crucial role in cell migration and angiogenic mechanisms [62]. FLNB also plays a role as a binding molecule along with ICAM-1 and co-localizes with it, forming a cluster. Reduced expression of FLNB protein results in lower expression of ICAM-1 and decreased lateral mobility, affecting trans endothelial transmigration [63,64,65]. The results show that the expression of the *FLNB* gene is down-regulated, leading to a significant change in the transcriptome expression of ICAM-1. These findings suggest that the *P. falciparum* parasite influences the downregulation of the *FLNB* gene, possibly affecting cellular transmigration.

The *TXNRD1* gene encodes thioredoxin reductase (TrxR), which is responsible for regulating oxidative damage and protecting cells [66]. TrxR regulates the activity of NF-kappa B and the cellular inflammatory response in vascular endothelial cells [67]. Additionally, the *TXNRD1* gene plays a role in detoxifying ROS and managing redox signaling, making it a potential factor in epilepsy, which is often caused by increased oxidative stress in the brain [68]. TrxR1 is one of the enzymes that may help protect against oxidative stress in the cortex and dopaminergic neurons, reducing brain damage during seizures [69]. The transcriptomic results showed a decreased expression of the *TXNRD1* gene, suggesting that oxidative stress may not be regulated effectively, leading to decreased neuroprotective effects. In a translational approach, seizures are a common symptom of severe malaria, suggesting that the *TXNRD1* gene may be involved in seizures associated with severe malaria.

Our results showed that the expression of heterogeneous receptors in HBMEC varies depending on the time of activation after being challenged by *P. falciparum* strains. Moreover, we have identified novel, previously undescribed endothelial genes related to CM as a result of parasitic activation of HBMEC. ICAM-1 is one of the most common receptors that mediates cytoadherence and is involved to the pathology of malaria [8,70], as well as ICAM-4 protein, which plays a role in host cell invasion by *P. falciparum* [71]. The cell adhesion molecules (CAMs) play a crucial role in maintaining endothelial integrity and regulating leukocyte migration [72]. Although molecules associated with parasite adhesion, such as integrin α3β1, VE-cadherin, JAM-A, JAM-B, and ICAM-2, were modulated, there was no statistically significant difference [73]. The involvement of Integrins αV and PECAM-1 in CM has been reported [74,75,76]; however, the significance of ICAM-3 and N-cadherin in malaria disease has not been established yet. The up-regulation of cxcl11 [73], RANTES/CCL5 [74,75], CXCR4 [76,77,78,79,80], eotaxin-2/CCL24 [81,82], MCP-1 [81,83,84], and CXCL5 [85,86] are associated with fever, leukocyte recruitment, and increased inflammation in malaria. Low levels of CCL28 were detected in individuals with malaria symptoms, which is a chemokine involved in regulating angiogenesis and nitric oxide [86]. The involvement of chemokines and cytokines in the pathogenesis of malaria was identified through a transcriptomic approach. Our work highlights and reinforces the crucial role of these inflammatory mediators in CM. The expression of ANGPTL4 as both a protein and gene was found to have increased in human brain endothelial cells that were stimulated with plasma from patients with *P. falciparum* malaria [85]. In cases of CM, changes occur in the coagulation process, causing up-regulation of factors such as coagulation factor III [22] and endothelin [87,88,89,90,91] which are linked to long-term cognitive impairment. Thrombospondin also plays a role in mediating the attachment of infected red blood cells to endothelial cells [92].The increased expression of vWF causes cerebral microangiopathy and increases the permeability of the BBB [93,94,95,96,97,98]. The absence of tropomodulin-2 has been linked to hyperactivity, learning deficits, behavior issues, and long-term memory problems [99]. A decrease in the expression of tropomodulin-2 was observed, and it suggests that the genes involved in coagulation play a crucial role in the coagulation cascade in cerebral malaria.

Our results support the involvement of MMPs [100,101,102,103,104] and TIMPs [91,101,102,103] in malaria pathogenesis. Moreover, our transcriptomic data revealed that the *P. falciparum* parasite modulates the expression of MMPs and TIMPs at the level of human endothelial cells.

In addition to transcriptomic and protein analysis of HBMEC, we conducted inflammatory studies on the secretomes of endothelial cells. Our findings are consistent with previous reports. Circulating proteins with high concentrations, such as TGF-β, TNF-α, IFN-γ, IL-2, IL-6, IL-8, IL-10, MCP-1, MIP-1β, Eotaxin, and IL-1, were identified in human cases of CM [81,105]. Conversely, a lower level of RANTES concentration was observed in severe human malaria patients [102,104,106,107]. We were able to reproduce these clinical observations in our in vitro study.

Our human brain organoids were successfully developed, as evidenced by immunofluorescence assays which showed normal reproducible development and maturation of the organoids. Our results indicate that iPSCs differentiate into brain organoids, forming neural tubes and regions with structures similar to the choroid plexus. TBR1 is involved in neuronal differentiation and is present in later stages of neuronal maturation, as well as postmitotic projection neurons, which are essential for normal brain development [108]. However, our findings showed that TBR1-positive cells were not detected in brain organoids.

During the developmental stages of brain organoids, it is common to observe a progressive organization with immature neurons predominantly concentrated in the outer region and fewer in the inner region, accompanied by a gradual thickening of the TUBB3+ cell layer [109], particularly in the cortical plate [110]. Our data showed positive cells for TUBB3, which aligns with the previously reported findings of a gradual thickening of the TUBB3+ cell layer in the outer layer of the organoids.

Brain organoids with 35 days of differentiation exhibit increased expression of MAP2, a marker of neural tissue, in the cortical plate (CP) [111]. Our immunofluorescence images revealed that MAP2 was present in a higher number of cells in the BO_MM condition, as evidenced by a clearer visual representation.

NESTIN is a recognized marker for neuronal progenitor cells in the adult brain and enables differentiation of neural progenitors from differentiated cells in the neural tube [112]. Brain organoids that have undergone 45 days of differentiation displayed radial glia (RG) cells, which were identified by the NESTIN marker and also displayed structures resembling cerebral ventricles [110]. Our data indicated the presence of positive cells for the NESTIN marker, as expected.

SOX2 is a marker for neural stem cells and is expressed in the cells of the neural tube and proliferating CNS progenitors. Brain organoids have functional human astrocytes as indicated in the literature [110,113,114]. Our data showed the presence of positive cells for SOX-2, but with a lower level of expression.

GFAP is a marker for astrocytes in the CNS and brain organoids that have matured for 100 days showed an increased level of positivity for GFAP, as described by Ref. [115]. Our results showed positive cells for GFAP, particularly in the outer layer of the organoids compared to the inner layer.

In brain organoids, the radial glia scaffold can be identified through the presence of the DCX marker, as described in the literature [111]. Our data showed positive cells for DCX in all conditions, with a higher expression of the marker in the outer layer of the brain organoids.

Caspase-3 is a marker of cell death and plays a crucial role in the development of brain injuries and neurodegenerative diseases, showing an increased profile [116,117]. Additionally, it is described that microcephaly brain organoids showed a significant increase in CASP3+ cells [118]. Our findings showed that all conditions had positive cells for CASP3 in brain organoids for all conditions, but without a higher concentration.

It has been established that ciliary function is essential in the development of the central nervous system in the human brain. Some brain cells, such as choroid plexus cells and ependymal cells facing the ventricles, have motile cilia. Furthermore, ciliopathies are linked to neurodevelopmental disorders and newborn problems, including mental retardation and structural deficits, as described in studies [119,120,121,122]. Human brain organoids can replicate microcephaly and have shown cilium disassembly, which contributes to microcephaly, as previously described [122]. Our results in brain organoids demonstrated a pattern of altered structural organization, which together with our transcriptomic data suggest an impaired neurological development. Additionally, the transcriptomic results indicated changes in the brain organoids related to the microtubule bundle, cilium organization, and extracellular organization. The changes identified by our transcriptomics at the biological, molecular, and cellular level can provide novel insights into the impact of activated BBB endothelial cells on the brain.

## 5. Conclusions

Our results offer an opportunity to enhance understanding of the underlying mechanisms of CM pathogenesis, which resembles the BBB endothelium in CM. Furthermore, the data presented in this study contributed to a better understanding of the injury mechanisms of CM following endothelial activation. These insights were obtained in a non-invasive manner through in vitro human endothelial and brain experimental models, allowing us to learn more about the physiological processes underlying CM. These models can serve as an important biological model to study brain development, complexity, and organization in the context of CM. We also highlight the use of brain organoids to gain insights into the neurological dysfunctions associated with CM. We have established a translational study that traces the effects of different *P. falciparum* strains on human brain endothelial activation up to their impact on brain organoids. Advancements in human brain organoid technologies, such as vascularization and co-culture techniques, and our results can support the future of in vitro CM research using human brain endothelial cells and human brain organoids as models of CM.

## Figures and Tables

**Figure 1 cells-12-00984-f001:**
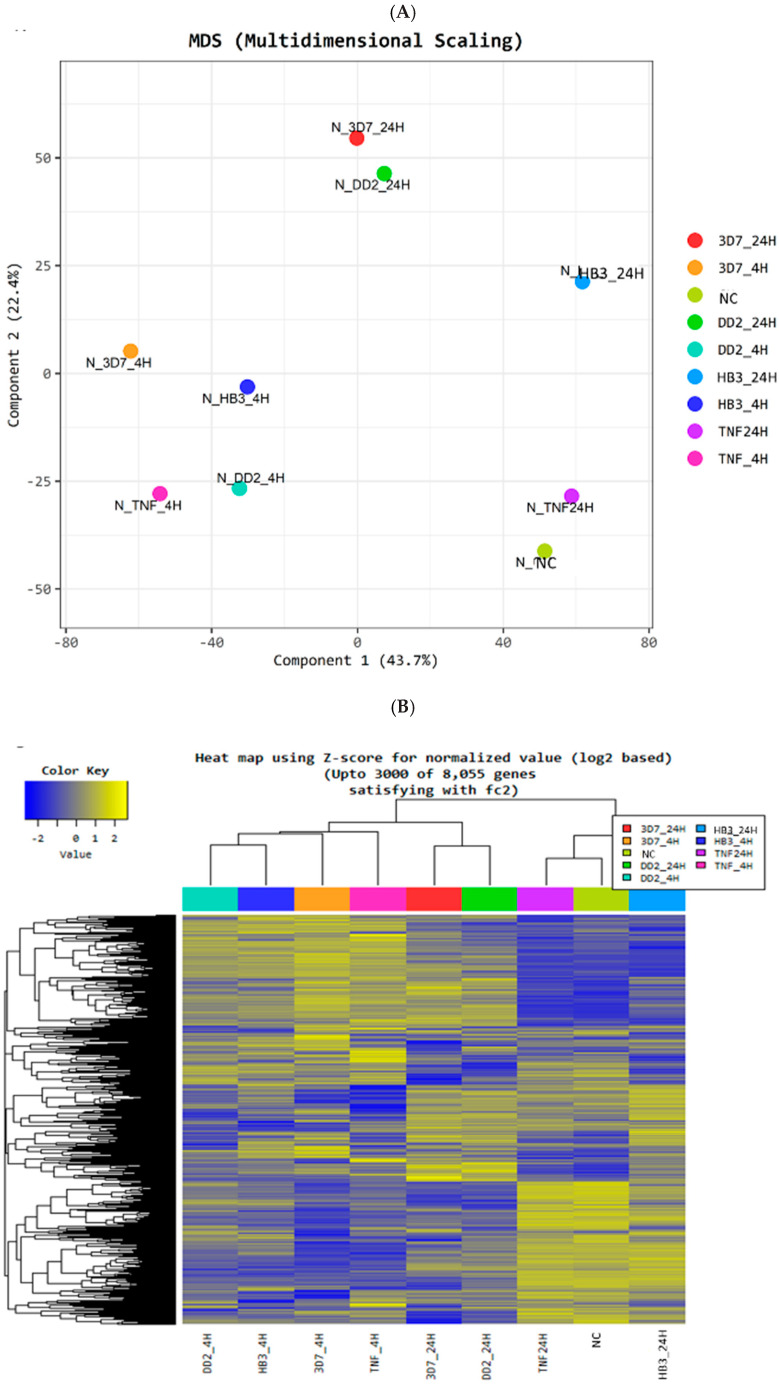
Data quality and data analysis description of HBMEC transcriptome. (**A**) Multidimensional Scaling Analysis: Using each sample’s normalized value, a 2D plot illustrates the variability of the total data. (**B**) Heatmap for DEG-Hierarchical Clustering: The similarity of genes and samples by expression level using Z-score for normalized value (log2 based) are clustered. Up to 3000 of 8055 genes from the significant list satisfied the |fc| ≥ 2. Each column represents a condition. Genes that were down-regulated are shown in blue, and up-regulated in yellow. (**C**) Significant UP and DOWN count by fold change shows up-regulated and down-regulated genes based on fold change of comparison pair. (**D**) Volume Plot between pooled samples for sample versus negative control (NC). Expression volume is the geometric mean of two groups’ expression levels. A volume plot is drawn to confirm the genes that show higher expression differences compared to the control according to the expression volume (*X*-axis: Volume, *Y*-axis: log2 Fold Change).

**Figure 2 cells-12-00984-f002:**
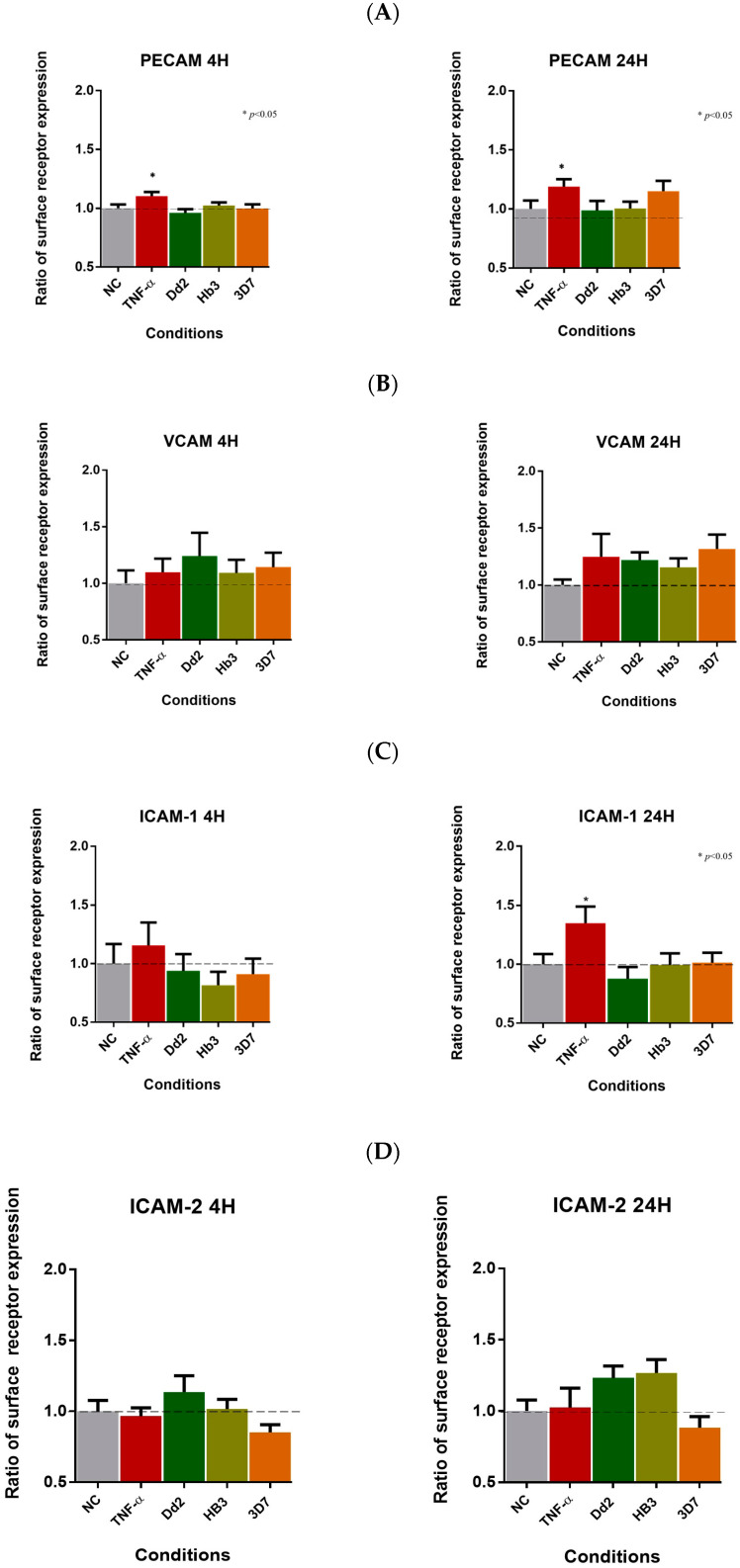
Cell receptor expression levels. Flow cytometry levels of receptors PECAM-1, VCAM-1, ICAM-1, EPCR, CD36, N-cadherin, JAM-A, Integrin alpha V beta 3, and ICAM-2 in HBMEC, when stimulated with Dd2, HB3, and 3D7 strains of *P. falciparum* parasites. Levels of surface expression of (**A**) PECAM-1, (**B**) VCAM-1, (**C**) ICAM-1, (**D**) ICAM-2, (**E**) CD36, (**F**) EPCR, (**G**) JAM-A, (**H**) N-cadherin, and (**I**) Integrin αVβ3, and expression on endothelial cells when in the presence of stimulation with TNF-α and *P. falciparum* parasites strains for 4 h and 24 h of activation, respectively. The ratio of receptor expression is in comparison to the negative control (NC) levels. The bar graphs show mean values and corresponding ± standard error of the mean (SEM) (*n* ≥ 3 independent experiments and 3 replicates per experiment). ****, *p* < 0.0001; ***, *p* < 0.001, **, *p* = 0.01; *, *p* < 0.05 (One-way ANOVA with Tukey’s posttest). Compilation of valuation of the levels of PECAM-1 (**J**), CD36 (**K**), JAM-A (**L**), and Integrin alpha V beta 3 (αVβ3) (**M**) receptor expression of HBMEC, when stimulated with Dd2, HB3, and 3D7 strains of *Plasmodium falciparum* parasites, and in the presence of TNF-α, between the time points 4 h and 24 h. The bar graphs show mean values and the corresponding ± standard error of the mean (SEM) (*n* ≥ 3 independent experiments and 3 replicates per experiment). ****, *p* < 0.0001; **, *p* = 0.01; *, *p* < 0.05 (Two-way ANOVA with Bonferroni test).

**Figure 3 cells-12-00984-f003:**
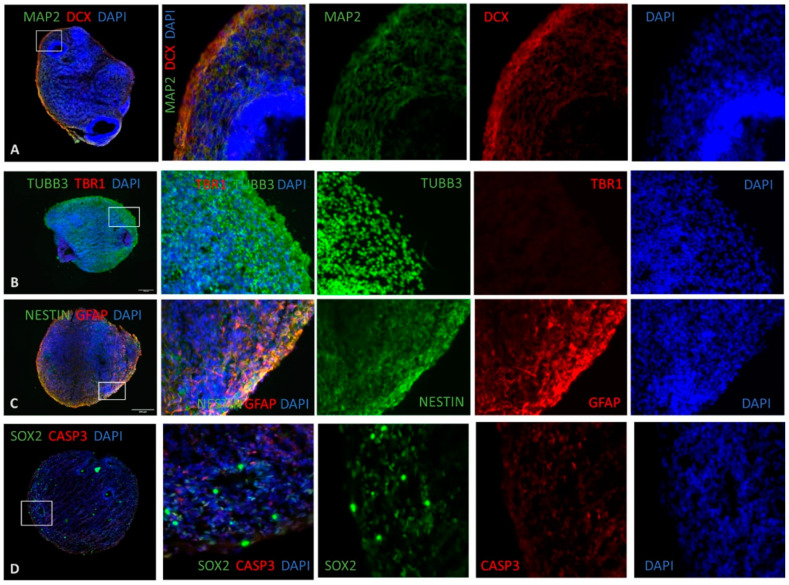
Differentiation of human cerebral organoids in vitro. Representative immunostaining of human cerebral organoids with 45 days post-differentiation, in cryosections with 10–15 μm. Scale bars: 200 μm. Nuclear staining with DAPI (blue). (**A**) Staining for the neuronal marker MAP2 (green) enlightens a superficial preplate, and for newborn neurons, marker doublecortin (DCX) (red). (**B**) Staining for cortical-layer neurons by tubulin beta 3 (TUBB3) (green) marker, showing a large cortical region. The pre-plate marker Tbr1 (red) for early-born neurons did not show any presence, thus revealing that early-born neurons were scarce and the brain organoids were mature. (**C**) Immunostaining of human cerebral organoids derived from iPSCs expressed for a neural stem cell marker (NESTIN) (green) at a low level and some glial fibrillary acidic protein (GFAP, human astrocytic marker) (red). (**D**) Immunostaining with neural progenitor cells (NPC) marker SOX2 for the ventricular zone (VZ) and apoptosis marker Caspase 3 (CASP3).

**Figure 4 cells-12-00984-f004:**
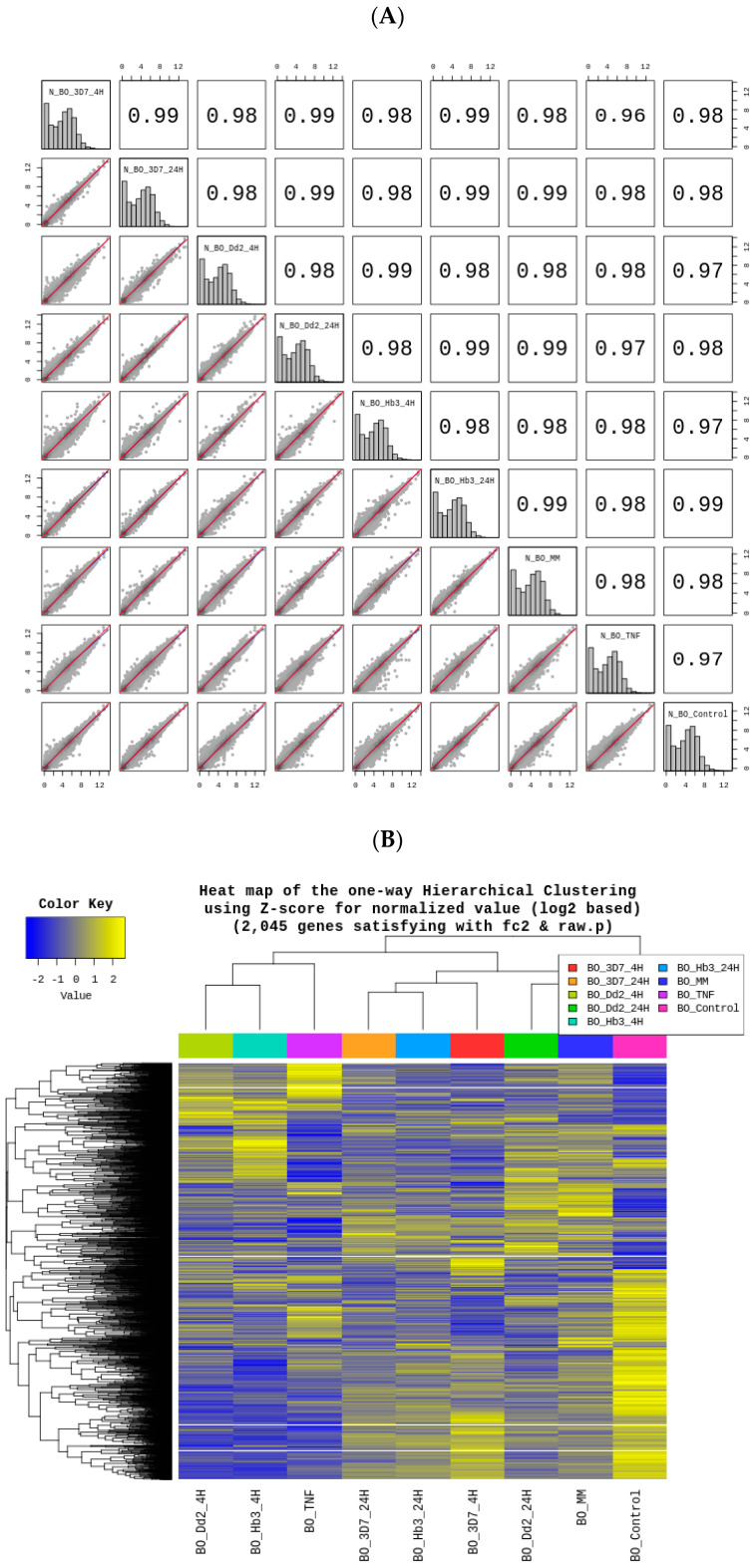
Data quality and data analysis of cerebral organoids transcriptome. (**A**) Reproducibility between total samples: The similarity between samples obtained through Pearson’s coefficient of the sample’s normalized value, for range: −1 ≤ r ≤ 1. (**B**) Heatmap for DEG-Hierarchical Clustering: Clusters the similarity of genes and samples by expression level using Z-score for normalized value (log2 based). A total of 2045 genes satisfy |F| ≥ 2 and *p*-value (*p* < 0.05) from the significant list. Each column represents a condition. Genes down-regulated are in blue and up-regulated are in yellow. (**C**) Multidimensional Scaling Analysis: Using each sample’s normalized value, the similarity between samples is graphically shown in a 2D plot to show the variability of the total data. (**D**) Significant UP and DOWN count by fold change (|F| ≥ 2) and *p*-value (*p* < 0.05) shows the number of up and down-regulated genes based on fold change of comparison pair.

**Figure 5 cells-12-00984-f005:**
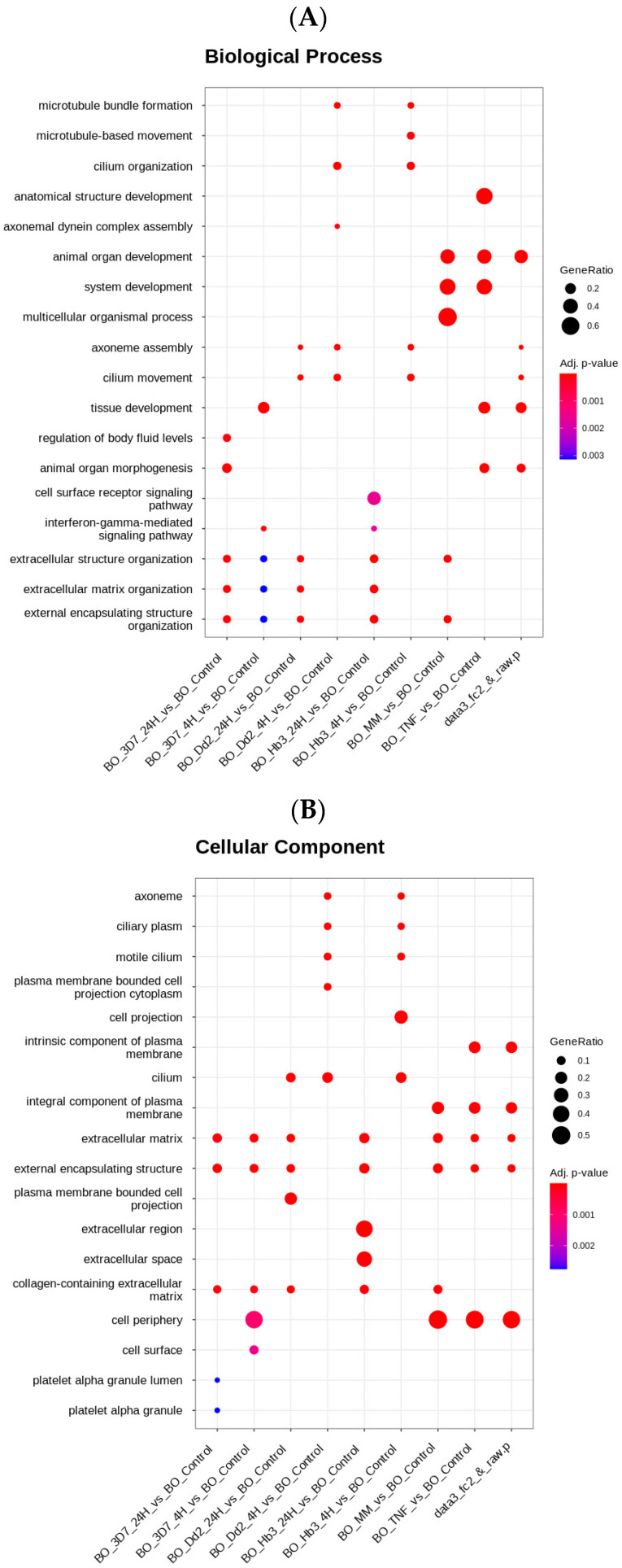
Gene Ontology (GO) from cerebral organoids transcriptomic. Dot plot shows significant Gene Ontology (GO). GO terms in each comparison were chosen if their adjusted *p*-value (adj. *p*-value) was less than 0.05. The top 10 GO terms are shown. The x-axis and y-axis were Gene Ratio (=intersection size/query size) and GO terms, respectively. Dot size was intersection size. Single-cell RNA sequencing (scRNA-seq) of cerebral organoids was performed after stimulation with secretome from P. falciparum-HBMEC. The transcriptomes analyzed were under several conditions, such as with TFN-α stimulation at time point from 4 h, which works as a positive control (BO_TNF), three different wild-type strains, 3D7, Dd2, HB3 of P. falciparum parasites induction, from 4 h (BO_3D7_4H, BO_Dd2_4H, BO_HB3_4H) and 24 h (BO_3D7_24H, BO_Dd2_24H, BO_HB3_24H), respectively. Cerebral organoids stimulated with the secretome from HBMEC without any stimulation named modified media (BO_MM). (**A**) Biological Processes (**B**) Cellular Component (**C**) Molecular function.

**Table 1 cells-12-00984-t001:** Protein microarrays exhibiting a significantly altered inflammatory profile. Expression of cytokines and chemokines in the secretome resulting from HBMEC-*P. falciparum* parasite stimulation and TNF-α [20 ug/mL] at time points 4 h and 24 h, compared with NC. In addition, fold-change value, *p*-value, and if proteins are increasing (↑) or decreasing (↓) were indicated.

*P. falciparum* Strain & Time-Point	Pro-Inflammatory	Anti-Inflammatory	*p*-Value	Fold-Change	Increase/Decrease
**3D7_4h**	RANTES		0.0001	0.2	↓
	IL-6	IL-6	0.0002	2.1	↑
	MCP-1		0.0001	2.1	↑
		IL-8	0.0006	2.7	↑
**HB3_4h**	RANTES		0.0001	0.2	↓
	IL-6	IL-6	0.0001	2.3	↑
**Dd2_4h**	RANTES		0.0001	0.3	↓
	IL-6	IL-6	0.0001	2.0	↑
**3D7_24h**	RANTES		0.0001	0.7	↓
	IL-6	IL-6	0.0001	2.5	↑
	MCP-1		0.0006	2.6	↑
		PDGF-BB	0.0022	2.6	↑
		GM-CSF	0.0001	4.4	↑
		TIMP-2	0.0005	2.4	↑
		IL-8	0.0101	2.7	↑
**HB3_24h**	RANTES		0.0105	0.7	↓
	IL-6	IL-6	0.0001	2.0	↑
	MCP-1		0.0006	2.4	↑
		PDGF-BB	0.0209	2.3	↑
		GM-CSF	0.0224	0.7	↑
		TIMP-2	0.0001	2.3	↑
**Dd2_24h**	IL-6	IL-6	0.0001	2.6	↑
	MCP-1		0.0001	2.8	↑
		PDGF-BB	0.0044	2.5	↑
		GM-CSF	0.0042	3.2	↑
		TIMP-2	0.0016	2.5	↑
		IL-8	0.0097	2.7	↑

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
