# Peer review of "Cerebral Malaria Model Applying Human Brain Organoids"

_cells, 2023, doi:10.3390/cells12070984_

Round 1

Reviewer 1 Report

The study by Silva-Pedrosa et al. describes the use of human brain organoids to investigate cerebral malaria. The area of investigation is timely and the use of human brain organoids promises to provide insights into the pathogenesis of cerebral malaria and allow identification of important biomarkers of disease. The study is of great interest. However, in the current form, the significance of the data is unclear and data are hard to evaluate because labels for the figures (A1, A3, A4, A5, A7 and A8) are very difficult to read. The transcriptome data needs a clearer explanation to determine if genes upregulated or down regulated in the human brain organoid model further our understanding of the pathogenesis of cerebral malaria in vivo.

1     1. The manuscript needs to be revised for organization, grammar and clarity. Labels for figures need to be revised. The abstract needs to concisely state the rationale for the study, the major approaches employed and the data obtained.

2      2. In the introduction, lines 53-68 belong in the discussion. Line 57 “…this activation is train specific…” needs to be revised. It is not clear from the introduction why the study was performed and what hypotheses were being tested.

3      3. In line 102, why refer to the media as the “secretome”? An explanation is required. See line 151 and following.

4      4. What is the rationale for the “secretome” conditions used in lines 198-210 and the controls used? What components determine activation and differentiation when half HBMEC culture is used with half P. falciparum culture? See Figure A5. Labels are very hard to read and the figure legend does not specify whether distribution of markers is in the same location within the boxed area of the panels in the figure. Ultrastructural data is required to confirm features and structures of development and differentiation of the organoids post-induction.

5      5. The types of multi-well plates used and the specific volumes of media and other reagents used should be consistently mentioned. See lines 240-241.

6      6. Frequent references to preceding methods for each method described is unnecessary in the description of methods.

7      7. Are the results in lines 271-272 from the current study or from previous studies?

8      8. Tabulate the results of receptor expression on HBMEC (lines 316-345) and other data (lines 348-400, 360-400 and 402-430) for clarity.

 9. It is not clear from the transcriptome data if expression of genes like COX1 and RNR2 have a role in blood brain barrier dysfunction in vivo.

1   10. Revise and correct “(Error! Reference source not found.)” and provide missing references.

1   11. The discussion needs to be revised and shortened to focus more on the discussion of the data from the current study and clearly identify the significance of the data obtained in the current study and an explanation for the data regarding differences in P. falciparum strain induced alteration of the transcriptome.

   12. A selection of the major data showing characterization of the human organoids and receptor expression should be provided in the main article rather that have all figures in a supplementary folder. Similarly, focus citations to the relevant topic of the study and reduce the number of articles cited in the bibliography.

Author Response

Dear Reviewer,

we appreciate your feedback and suggested changes that significantly improve this article. Highlighted points were added into the manuscript. The abstract and introduction were fully restructured as suggested.

The study by Silva-Pedrosa et al. describes the use of human brain organoids to investigate cerebral malaria. The area of investigation is timely and the use of human brain organoids promises to provide insights into the pathogenesis of cerebral malaria and allow identification of important biomarkers of disease. The study is of great interest.

However, in the current form, the significance of the data is unclear and data are hard to evaluate because labels for the figures (A1, A3, A4, A5, A7 and A8) are very difficult to read. The transcriptome data needs a clearer explanation to determine if genes upregulated or down regulated in the human brain organoid model further our understanding of the pathogenesis of cerebral malaria in vivo.

  • The manuscript needs to be revised for organization, grammar and clarity. Labels for figures need to be revised. The abstract needs to concisely state the rationale for the study, the major approaches employed and the data obtained. A: The organization, grammar and the English were rectified.
  • In the introduction, lines 53-68 belong in the discussion. Line 57 “…this activation is train specific…” needs to be revised. It is not clear from the introduction why the study was performed and what hypotheses were being tested. A: We added our major goals in the abstract.
  • In line 102, why refer to the media as the “secretome”? An explanation is required. See line 151 and following. A: The media is called “secretome”, because refers to Plasmodium falciparum activated HBMEC which generates a secretome response containing the set of specific proteins response.
  • What is the rationale for the “secretome” conditions used in lines 198-210 and the controls used? A: We used a short-time (4H) and a long-time of activation (24H) of endothelial cells by P. falciparum strains to understand if the endothelial cells response change over the time and the parasite strain. The secretome from HBMEC-P. falciparum were used to stimulate the brain organoids to better understand what changes occur in the brain following the endothelial activation. Since the HBMEC are a central element of the microvasculature that forms the blood-brain barrier and shields the brain against toxins and immune cells via paracellular, transcellular, transporter, and extracellular matrix proteins.
  • What components determine activation and differentiation when half HBMEC culture is used with half falciparum culture? A: The culture mediums used respective endothelial and parasite culture mediums, where the environment and nutrients need are respected for both cell growth and parasite maintenance. Negative control was culture with same conditions showing homeostatic natural behaviour. Moreover, our results showed a specific activation induced by P. falciparum in HBMEC, including when our positive control (TNF-α) didn’t induce activation of endothelial cells.

See Figure A5. Labels are very hard to read and the figure legend does not specify whether distribution of markers is in the same location within the boxed area of the panels in the figure. Ultrastructural data is required to confirm features and structures of development and differentiation of the organoids post-induction. A: The legend size it was changed. The information about boxed area was add, that represents heterogeneous regions (red square) and choroid plexus architecture and neural tubes (white square).

  • The types of multi-well plates used and the specific volumes of media and other reagents used should be consistently mentioned. See lines 240-241. A: The information it was added.
  • Frequent references to preceding methods for each method described is unnecessary in the description of methods. A: It was corrected the references in methods, only remain the references that aggregate information about the origin.
  • Are the results in lines 271-272 from the current study or from previous studies? A: From the current study.
  • Tabulate the results of receptor expression on HBMEC (lines 316-345) and other data (lines 348-400, 360-400 and 402-430) for clarity. A. The results were tabulated and added to supplementary appendix.
  • It is not clear from the transcriptome data if expression of genes like COX1 and RNR2 have a role in blood brain barrier dysfunction in vivo. A: Our work demonstrated that COX-1 was one of the genes more expressed in human brain microvascular endothelial cells when they are activated by P. falciparum. This not only establishes a possible relation between malaria's effect on COX-1 gene expression but the COX-2 and COX-3 genes.
  • Revise and correct “(Error! Reference source not found.)” and provide missing references. A: This happened because the reference to the figure is in automatic mood, and after I tested, in different laptops showed as error. The alterations were made.
  • The discussion needs to be revised and shortened to focus more on the discussion of the data from the current study and clearly identify the significance of the data obtained in the current study and an explanation for the data regarding differences in P. falciparum strain induced alteration of the transcriptome. A: The introduction, abstract, discussion and conclusion have undergone profound changes. A more focused discussion was established between the results obtained in our study and what is described in the literature.
  • A selection of the major data showing characterization of the human organoids and receptor expression should be provided in the main article rather that have all figures in a supplementary folder. Similarly, focus citations to the relevant topic of the study and reduce the number of articles cited in the bibliography. A: The data has been added to the main article. As there are no studies of brain organoids to understand cerebral malaria, it was used for discussion and correlation the existing literature. However, the bibliographical references have undergone changes in order to be more in line with the main theme.

Reviewer 2 Report

Abstract and Introduction:

The last keyword in the abstract can be replaced with a specific word. (Line 24-25)

WHO Malaria Report 2022 is available, and it is suggested that authors update the statistics and citations accordingly. (Lines 28-30)

The patients usually claim symptoms. Hence, it is recommended to replace the term clinical symptoms with “clinical signs.” (Line 30)

Malaria itself is a disease. Hence, there is no need to use the phrase “Malaria Disease”. (Line 32)

How can the authors claim that the etiology of the CM is unclear? Please provide the latest reference. While in the coming sentence, they discussed it. Additionally, use the word “and” between pathogenesis and etiology. (Lie 34)

Please clarify endothelium or epithelium. (Line 38)

Explain the word “train”. (Line 57)

One of the significant aspects of the current manuscript is related to the interplay between endothelial cells and the Malarial parasite. There are plenty of research articles available on this topic. However, the authors have not cited any such study in the introduction. Please include this part and improve the introduction.

The results can be described or discussed in the results and discussion section. Please refrain from stating results in the introduction section. (Lines 53-62)

Authors must follow the journal’s recommended reference style.

Materials & Methods:

What is Plasmodium falciparum Dd2, and HB3 etc.?

Please provide the location of the Malaria Research Center. (Line 73)

How were the human red blood cells sourced?

Rewrite the sentence to reduce confusion. (Line 93-95)

What are the parasite lines? (Line 97)

Do correct the unit throughout the manuscript. (Line 104)

What was the coating agent? (Line 106)

Justify the washing of cells with FACS. (Line 113)

Correct the name of the statistical test. (Line 126, 172)

The sentence needs to be more straightforward. (Line 133-135)

The sentence is incomplete (Line 158-159)

Kindly use the proper symbols (Line 206-207)

Results:

What was the hypothesis to verify several genes? (Lines 288-294)

Provide the complete form of the abbreviation “iRBSs” (Line 299)

The organism’s name needs to be italicized. (Line 316).

Provide the reference to the claim. (Line 349-351).

Remove the “Error!”. (Lines 360-400)

It is suggested to avoid writing tiny paragraphs. Please rectify it. (Lines 401-430).

The authors cultured their organoids for more than 40 days (post-differentiation culture). How can a brain organoid mature in such a short period? Please explain (Lines 458-460)

It is also recommended that the authors must be consistent in using the term brain organoid vs. cerebral organoid.

Discussion:

The discussion must compare the scientific findings with the already published literature. Several sentences in the discussion sections look like extensions of the results. In several places, the font size is inconsistent.

The paragraph explaining the general benefits of organoids is unnecessary. Please omit it or move it to the introduction (if necessary). (Line 579-584)

It would be great if the authors made a separate section, “Conclusion: out of the discussion.

Figures:

Font sizes are too small, and the font style differs from the recommended " Cells " style.

What is the scale of Figures A5 and A6?

Some figures can be moved to the supplementary information section.

Author Response

Dear Reviewer, we appreciate your feedback and suggested changes that greatly improve our manuscript. The highlighted points were added into the manuscript. The abstract and introduction were fully rewritten as suggested.

Abstract and Introduction:

  • The last keyword in the abstract can be replaced with a specific word. (Line 24-25). A: it was replaced by HBMEC activation.
  • WHO Malaria Report 2022 is available, and it is suggested that authors update the statistics and citations accordingly. (Lines 28-30). A: The information has been replaced from WHO Malaria Report 2021 to the WHO Malaria Report 2022.
  • The patients usually claim symptoms. Hence, it is recommended to replace the term clinical symptoms with “clinical signs.” (Line 30)
  • Malaria itself is a disease. Hence, there is no need to use the phrase “Malaria Disease”. (Line 32)
  • How can the authors claim that the etiology of the CM is unclear? Please provide the latest reference. While in the coming sentence, they discussed it. A: the term "etiology” is not correctly used, we wanted to say is “molecular etiology”. The precise mechanisms underlying the interplay between human brain endothelial cells and the malarial parasite are still not fully understood.
  •  
  • Additionally, use the word “and” between pathogenesis and etiology. (Lie 34)
  • Please clarify endothelium or epithelium. (Line 38) A: Epithelial cells with particular functions are known as endothelial cells. The fundamental distinction between epithelial and endothelial cells is that while endothelial cells line the internal surfaces of the circulatory system's components, epithelial cells only line the body's internal surfaces.
  • Explain the word “train”. (Line 57) A: It was a typo error, which is now changed.
  • The results can be described or discussed in the results and discussion section. Please refrain from stating results in the introduction section. (Lines 53-62) A: the most descriptive results were taken from the introduction.
  • One of the significant aspects of the current manuscript is related to the interplay between endothelial cells and the Malarial parasite. There are plenty of research articles available on this topic. However, the authors have not cited any such study in the introduction. Please include this part and improve the introduction. A: The introduction, abstract, discussion and conclusion have undergone profound changes. Also, more information about the interplay between endothelial cells and the malarial parasite it was added.
  • Authors must follow the journal’s recommended reference style.

Materials & Methods:

  • What is Plasmodium falciparum Dd2, and HB3 etc.? A. There are falciparum strains from different geographic origins and phenotypes. Changed in the manuscript.
  • Please provide the location of the Malaria Research Center. (Line 73) A: Manassas, VA. Provided now in the manuscript.
  • How were the human red blood cells sourced? A: using a magnetic cell separation, AutoMACS (Miltenyi Biotec), according to the manufacturer protocol using the program PMalaria.
  • Rewrite the sentence to reduce confusion. (Line 93-95)
  • What are the parasite lines? (Line 97) A: The parasite strains. Changed in the manuscript.
  • Do correct the unit throughout the manuscript. (Line 104). A: It was altered.
  • What was the coating agent? (Line 106) A: Surface Treatment, the SPL company doesn’t refer which coating agent it uses. However, it was added the
  • Justify the washing of cells with FACS. (Line 113).A: Suitable buffer is isotonic and buffered to neutrality, will cushion the cells against damage during centrifugation, block non-specific staining, prevent capping of bound antibody. The FACS buffer contains sodium azide as preservative and animal serum proteins (FBS/BSA) to help minimize non-specific binding of antibodies. The addition of EDTA prevents cell to cell adhesion and clumping, since we were analyzing cell surface staining (doi: 1016/j.xpro.2020.100270)
  • The sentence needs to be more straightforward. (Line 133-135)
  • Kindly use the proper symbols (Line 206-207)
  • The sentence is incomplete (Line 158-159)
  • Correct the name of the statistical test. (Line 126, 172)

Results:

  • What was the hypothesis to verify several genes? (Lines 288-294). A: The goal of studying genes from endothelial cells activated by Plasmodium falciparum is to understand the molecular mechanisms involved in the host response to the parasite's infection. The study of different genes is intended to provide a broader understanding of key genes and pathways involved in the host's defense against the parasite, providing insights into the general principles of host-pathogen interactions.
  • Provide the complete form of the abbreviation “iRBSs” (Line 299)
  • The organism’s name needs to be italicized. (Line 316).
  • Provide the reference to the claim. (Line 349-351).
  • Remove the “Error!”. (Lines 360-400). A: This happened because the reference to the figure is in automatic mood, and after I tested, in different laptops showed as error. The alterations were made.
  • It is suggested to avoid writing tiny paragraphs. Please rectify it. (Lines 401-430).
  • The authors cultured their organoids for more than 40 days (post-differentiation culture). How can a brain organoid mature in such a short period? Please explain (Lines 458-460) Brain organoids can reach maturity within 40 days, this might depend on the specific conditions and methods used in the lab. For example, can be obtained neural maturation (https://doi.org/10.1016/j.stemcr.2021.06.011) (10.15283/ijsc21195) and glial cells in brain organoids with 40 days (https://doi.org/10.1038/s41598-022-16369-y)
  • It is also recommended that the authors must be consistent in using the term brain organoid vs. cerebral organoid. It was changed for brain organoids term.

  • Discussion: A more focused discussion was established between the results obtained in our study and what is described in the literature
  • The discussion must compare the scientific findings with the already published literature.
  • Several sentences in the discussion sections look like extensions of the results. In several places, the font size is inconsistent.
  • The paragraph explaining the general benefits of organoids is unnecessary. Please omit it or move it to the introduction (if necessary). (Line 579-584). A: It was removed this description part.
  • It would be great if the authors made a separate section, “Conclusion: out of the discussion. It was separated the discussion from conclusion.

Figures:

  • Font sizes are too small, and the font style differs from the recommended " Cells " style.
  • What is the scale of Figures A5 and A6? A:The scale bar it is 200 micrometers (200 μm).

Some figures can be moved to the supplementary information section. A: We believe that the images in the article are important to convey the necessary information. However, we would like to know your suggestion on which images to move for supplemental data to meet the intended purpose.

Round 2

Reviewer 1 Report

The authors have responded to the questions asked. However, the manuscript still needs to be corrected for grammar. Several passages and headings need to be revised for clarity throughout the manuscript.

A few examples are shown below. Corrections are needed throughout the manuscript.

Line 171: Revise the first sentence.

Lines 268-269: “Per experiment were pooled 5 organoids,…”

Line 387: “Activation of receptors expression”

Line 484: “At 4 h, it was observed a significant…”

Line 587: “After transcriptomic profiles been normalized (Fig.A8) was possible to verity…”

Lines 803-804: “Furthermore, our results contributed to a better understanding of…”

Author Response

Dear Reviewer

Thank you for taking the time to review our manuscript. We appreciate your thoughtful comments and feedback on our work.

We have made several changes to the manuscript, including suggested ones, such as revising the headings and passages throughout the manuscript for clarity.

In conclusion, we have carefully considered your feedback and have made the necessary revisions to improve the quality of our manuscript. We believe that the changes we have made have strengthened the overall argument and presentation of our research.

Again, thank you for your time and effort in reviewing our manuscript. Your insights have been invaluable, and we look forward to the possibility of seeing our revised manuscript published in Cells.

Sincerely

Ana Rita Pedrosa

Reviewer 2 Report

1. Authors must replace figure 1 with high-resolution images of the graphs etc. The same goes to figure 2. Additionally, the font size in figure 2 is too small.

2. What are (BO_3D7_4H, BO_Dd2_4H, 259 BO_HB3_4H) and 24h (BO_3D7_24H, BO_Dd2_24H, BO_HB3_24H), what is this coding for? Authors can use better labeling numbers/names. Line 258-259

3. In table 1, hours are labeled as "H," while in the legend, it is donated as "h".

Author Response

Dear Reviewer,

Your comments and suggestions have greatly improved the quality of our work. I am pleased to inform you that we have made the requested changes and revisions to our manuscript based on your feedback.

Authors must replace figure 1 with high-resolution images of the graphs etc. The same goes to figure 2. Additionally, the font size in figure 2 is too small. A: In response to your comment regarding the figures, we have replaced some of the low-resolution images with high-quality, high-resolution versions in the manuscript. This has significantly improved the clarity and detail of the figures and will provide a better experience for readers reviewing the manuscript. Regarding the images in the document, we understand your concern about the quality of some of them. Unfortunately, due to the size and complexity of the document, it would be too heavy to change all the images at this stage. However, we have compiled higher quality versions of these images in a separate file, which we can submit along with the document.

  1. What are (BO_3D7_4H, BO_Dd2_4H, 259 BO_HB3_4H) and 24h (BO_3D7_24H, BO_Dd2_24H, BO_HB3_24H), what is this coding for? Authors can use better labeling numbers/names. Line 258-259. A: We would like to address your comments regarding the labeling of our coding for the different experimental conditions of challenged brain organoids (BO). In our study, we have labeled the different types of challenges as Dd2, 3D7, and HB3. Moreover, changing the name of figures leads to a loss of image quality.

We understand that the labeling may seem lengthy, but we have done so to facilitate more effective reading and identification of the conditions by the reader.

  1. In table 1, hours are labeled as "H," while in the legend, it is donated as "h". A: Additionally, we have made the requested changes to the label of Table 1, ensuring that it is now accurate and consistent with the content presented in the table.

We believe that these changes have further strengthened the manuscript and improved its overall quality. Once again, we are grateful for your feedback and input. Your expertise and attention to detail have helped us to produce a better manuscript, and we hope that our revisions meet with your approval.

Sincerely,

Rita Pedrosa
